# New directions in the search for dark matter

**Surjeet Rajendran**

Department of Physics & Astronomy, The Johns Hopkins University,
Baltimore, MD 21218, USA

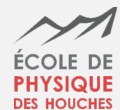

*Part of the Dark Matter*
*Session 118 of the Les Houches School, July 2021*
*published in the Les Houches Lecture Notes Series*

## Abstract

The identification of the nature of dark matter is one of the most important problems confronting particle physics. Current observational constraints permit the mass of the dark matter to range from $10^{-22}$ eV - $10^{48}$ GeV. Given the weak nature of these bounds and the ease with which dark matter models can be constructed, it is clear that the problem can only be solved experimentally. In these lectures, I discuss methods to experimentally probe a wide range of dark matter candidates.

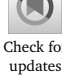
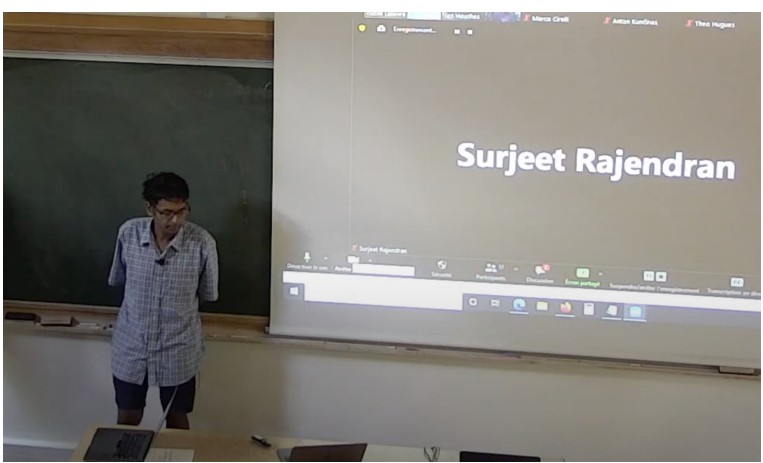

## 1 Introduction

The existence of dark matter proves that there is physics beyond the standard model. But, other than its existence, observational limits on its properties are extremely weak. Let us quickly review these observational constraints.

We know that the majority of the dark matter must have a mass less than about $\sim 10^{48}$ GeV [17]. This constraint arises from the observed absence of gravitational lensing that would be caused by a localized dark matter object. If it is a boson, its mass must be larger than $\sim 10^{-22}$ eV [18] while for a fermion the lower limit is closer to $\sim 10$ eV [26]. These limits arise from the fact that the dark matter fits inside the galaxy. A boson that is lighter than $\sim 10^{-22}$ eV will have too large a de Broglie wave-length to be confined within the galaxy while a fermion lighter than $\sim 10$ eV would have too large a number density to fit inside the galaxy while obeying the Pauli exclusion principle. The self-interactions of the dark matter need to be low enough that there is less than $\mathcal{O}(1)$ scattering in the dark matter during the time scale over which galaxy clusters merge [24]. Further, the dark matter is a cold pressure-less gas during the time of the cosmic microwave background [9]. Long range forces in the dark sector are also dominated by gravitation [1].

Unfortunately, these observational facts are not terribly constraining - it is easy to construct a wide variety of viable theoretical models that span the entirety of this observationally allowed parameter space. While every theorist on the planet has their favorite model of dark matter emerging naturally from their own undoubtedly well motivated theories, there is no doubt that if any dark matter experiment actually detects any kind of dark matter, every theorist on the planet will have no difficulty in coming up with a well motivated theory that completely explains the data. At this time, there is tremendous observational evidence to support this point of view [1].

---

[1] An experimentalist can even occasionally induce theorists to write serious theory papers [2] by making truthful but (willfully) mis-interpretable statements about data in their experiment.

Moreover, all of these constraints apply to the majority of the dark matter - the properties of sub-components (at the level of a few - 10 percent) are completely unconstrained. Note that there is nothing "wrong" about probing a sub-component of dark matter - these sub-components can arise naturally as cosmological relics and their presence may be the only way to detect some other sector of particle physics. As an example, the axion is a theoretically well motivated particle and in many cosmological contexts, there is a natural cosmic abundance of this particle. While it may easily be all of the dark matter, it is also not unusual for it to have a smaller cosmological abundance. Discovering such a cosmic sub-component would offer a direct way to discover the ultra-violet physics that produced such a particle. Another example is the cosmic neutrino background itself - the $C\nu B$ is a sub-component of the dark matter and just because it is a sub-component, it does not mean that detecting it is in any sense less interesting.

Given the vastness of this parameter space, how can we hope to make progress? When confronted with this vastness, there is a human tendency to artificially restrict it by focusing on "theoretically well motivated" dark matter - in this context, "theoretically well motivated" means particles that theorists have already written down for some other reason. While it is certainly possible that the existence of dark matter may be tied to the solution to some other problem in particle physics, such a connection is not a logical requirement. It is a fantasy to think that the particle spectrum of the world can be figured out entirely from first principles. I have not come across a physicist who has convinced me that their refined sense of theoretical insight would have allowed them to figure out (without experimental input) that the Standard Model is a $SU(3) \times SU(2) \times U(1)$ gauge theory with the $SU(3)$ confined at low energies, the $SU(2) \times U(1)$ broken in a weird way leaving an unbroken $U(1)$, with three generations of quarks and leptons that have hierarchial yukawa couplings with only the top quark possessing a naturally large yukawa coupling while also containing nearly massless neutrinos and a highly fine tuned Higgs boson. Our job as physicists is to discover what nature actually is rather than attempt to constrain it from the armchair.

These lectures are presented from that perspective: what are observational ways in which we can constrain the properties of dark matter? The vastness of the parameter space implies that we need to think of generic strategies that can probe a large class of signatures as opposed to focusing on specific predictions from specific theories. The development of this strategy requires theoretical input - whatever the dark matter is, it is highly likely that it is some particle that obeys the normal rules of quantum field theory. Field theory restricts the possible class of signatures that a particle might yield and the job of the theorist is to identify these classes of signatures and devise suitable experimental tests to detect these signatures.

I will divide these lectures into 4 parts, corresponding to 4 major classes of dark matter parameter space where progress seems possible. The first, discussed in section 2, deals with ultra-light dark matter. This describes bosonic dark matter particles whose mass can be as small as $\sim 10^{-22}$ eV ranging all the way to particles with masses close to $\sim$ meV. The second, discussed in section 3 describes dark matter that is light *i.e.* in the mass range $\sim$ meV - GeV. The third, discussed in section 4 refers to conventional heavy dark matter with mass above a GeV that would still go through a terrestrial scale detector in a year. The last section, section 5, discusses ways to probe super heavy dark matter, going all the way from $\sim 10^{16}$ GeV dark matter to planetary mass objects with a mass $\sim 10^{48}$ GeV. The boundaries between various parts of this parameter space are somewhat hazy and these lectures will largely be based on my own research into these topics. This is simply due to my own laziness, as I have material prepared on my own work. As you no doubt know, the development of techniques to detect dark matter is a very active area of research and there are a number of folks who have made insightful contributions to this area. The material presented in these lectures is not meant to be an exhaustive discussion of these efforts - it is simply meant to provide an insight into how

the problems of dark matter detection can be viewed with a strategic lens. Specifically, in each of these topics, we will ask the following questions: what is it that makes the detection of this class of dark matter difficult? What aspects of this class of dark matter can we leverage to overcome these limitations?

## 2 Ultra-Light Dark Matter

Bosonic dark matter in the mass range $\sim 10^{-22}$ eV - $\sim$ meV is generally called ultra-light dark matter or wave-like dark matter distinguishing it from heavier dark matter candidates that are supposed to be "particle" like. This sobriquet, while somewhat silly from the point of view of quantum mechanics, is somewhat appropriate - when the bosons are this light and they constitute the majority of the dark matter, the number density of the corresponding bosonic field is large enough that the modes of the quantum field that are populated by the dark matter have a large occupation number, effectively allowing the system to be described effectively as a classical "wave-like" field. This classical "wave-like" nature of the field will be important in detecting this kind of dark matter.

The principal difficulty in detecting ultra-light dark matter is that each dark matter particle has a very low kinetic energy since its mass is so light - thus if we were to come up with detection techniques that relied on the energy deposition of single particles it is likely that we will come up empty handed in trying to find these kinds of dark matter particles. Instead, we will leverage the large number density of the field and look for coherent effects caused by this field. In a sense, the idea of detecting this kind of dark matter is similar to trying to detect wind in a conventional human context. When we try to detect wind, the kind of wind gauge we build looks for the coherent effect of the wind pushing some kind of vane or gear (like in a windmill) as opposed to devising a fancy detector that looks for the energy deposition of a single wind molecule. Much like how we view the coherent effects of the wind as arising due to a fluid field, we can similarly think of the coherent effects of the dark matter as arising from the detectable effects of a classical field.

Since we want to leverage the properties of this classical field, we need to understand what this classical field looks like in the galaxy today. Now there are a variety of ways of producing ultra-light dark matter particles - they can for example be produced as spatially homogeneous fields during inflation, emitted as classical radiation from topological defects (such as strings) in the early universe or emerge as quantum fluctuations from inflation itself. No matter what their initial production mechanism, if they are to be the dark matter of the universe they would have seeded the growth of structure in the universe and have collapsed into galaxies today. What is the state of this classical field in the galaxy today? Of course, it is practically impossible to calculate the exact classical field today even if we are given some simple initial conditions due to the extra-ordinary complexity of the dynamical evolution involved in the formation of galaxies. For our purposes, we will simply regard this classical field as being completely random.

Now, even though this is a random classical field, we still know some properties about it. First, we know that the dark matter is cold - *i.e.* it is a non-relativistic system. So if the dark matter was some scalar $\phi$, we can describe it as an oscillating scalar field (think of how a photon is described by an oscillating electromagnetic field). The oscillation will occur at the energy of this particle - for a non-relativistic particle this is dominated by the rest mass of the system. Thus, we think of the dark matter as basically being an oscillating field $\phi_0 \cos(m_\phi t)$ where the amplitude $\phi_0$ is related to the energy density of the field as $m_\phi^2 \phi_0^2 \sim \rho_{DM}$ where $\rho_{DM}$ is the local dark matter density. This description has so far ignored the spatial profile of the field - it describes a completely spatially homogeneous field in the galaxy. In the galaxy,

we do not expect homogeneity - the field will have random inhomogeneities. Even though the inhomogeneities are random, we still define a correlation length for this field *i.e.* given the value of $\phi(x)$ of the classical (scalar) field $\phi$ at the point $x$, how far do we have to go before the field value is $\mathcal{O}(1)$ different?

Thinking about this problem in Fourier space, notice that changing the value of the field in position corresponds to the field possessing momentum. Thus the distance we need to travel before the field value is $\mathcal{O}(1)$ different is $\sim \frac{1}{m_\phi v}$ where $m_\phi$ is the mass of the boson and $v \sim 10^{-3}$ is the virial velocity of the dark matter in the galaxy. No matter what the state of the dark matter in the galaxy, we are guaranteed this minimal correlation length - simply because a shorter correlation length would correspond to a larger velocity for the particle and those particles will not be gravitationally bound to the galaxy and be the dark matter in the galaxy. In the following, we will devise various experiments to measure the coherent effects of the classical field $\phi$. For an experiment, we care not just about the correlation length of the field but the coherence time *i.e.* how long can an experiment sit at a point and measure the value of the field before this value changes by $\mathcal{O}(1)$? Now the relative velocity of the experiment and the dark matter is also $v$ and thus this coherence time is $\sim \frac{1}{m_\phi v^2}$. We can also think of this coherence time as arising due to the kinetic energy of the dark matter in the galaxy - if the dark matter did not have a kinetic energy, it would be a cold condensate that is at an energy equal to its mass $m_\phi$. The kinetic energy spreads this by $\sim m_\phi v^2$. Now since the kinetic energy is random, the time scale for phases associated with the kinetic energy to change by $\mathcal{O}(1)$ is $\sim \frac{1}{m_\phi v^2}$.

Thus, the problem of detecting ultra-light dark matter has been reduced to the following question: how can we detect an oscillating classical field that is oscillating at a frequency $m_\phi$ with a coherence time $\sim \frac{1}{m_\phi v^2}$? Now, this is where the story gets interesting - take for instance a particle that has a mass $m_\phi \sim$ MHz. We would expect this particle to give rise to an oscillating field that oscillates at a MHz frequency with the oscillation remaining coherent for $\sim \frac{1}{v^2} \sim 10^6$ periods $\sim 1$ s (for MHz frequency particle). These frequency and time scales are experimentally interesting since we know to create experimental devices that can respond at MHz frequencies and we are able to sit and make measuring devices that are able to acquire signals for time scales $\sim 1$ s.

Thus, instead of trying to detect the energy deposited by a single particle, we will try to detect these oscillating fields that oscillate at the unknown mass $m_\phi$ of the dark matter with a coherence time $\sim \frac{1}{m_\phi v^2} \sim \frac{10^6}{m_\phi}$. The fact that the oscillations of the field are coherent for $\sim 10^6$ periods implies that one can conceive of resonant schemes that will boost the dark matter signal.

What sorts of ultra-light bosons are interesting to us? We can put in a little bit of theory prejudice and say that if the ultra-light boson is going to be this light and still interacting with the standard model, we can apply considerations of technical naturalness *i.e.* we demand that the interactions of the boson possess some kind of symmetry so that the mass of the boson is protected from radiative corrections. When protected by symmetry, the dominant interactions between the boson and the standard model are restricted [14]. Let us list these possible interactions. For a scalar $\phi$, these interactions are:

$$\frac{\phi}{f_\phi} F\tilde{F}, \frac{\phi}{f_\phi} G\tilde{G}, \frac{\partial_\mu \phi}{f_\phi} \bar{\Psi}\gamma^\mu \gamma_5 \Psi, g\phi h^2. \tag{1}$$

The first operator in (1) is the coupling of an axion-like-particle to electromagnetism (field strength $F$) while the second couples $\phi$ to Quantum Chromodynamics (QCD) (field strength $G$) and is the defining coupling of the QCD axion (*i.e.* $\phi$ gets a mass from QCD instantons and if this scalar receives no other mass contributions, the dynamical evolution of this scalar to its

minimum solves the strong CP problem). The third operator in (1) is a derivative coupling of $\phi$ to the axial current of standard model fermions ($\Psi$) while the last is a higgs ($h$) portal coupling that has recently found a measure of popularity in relaxion models where such a scalar can solve the hierarchy problem via dynamical evolution.

For a vector boson $B_\mu$, the following interactions are possible:

$$\epsilon H_{\mu\nu}F^{\mu\nu}, \frac{H_{\mu\nu}}{\Lambda}\bar{\Psi}\sigma^{\mu\nu}\Psi, \frac{H_{\mu\nu}}{\Lambda}\bar{\Psi}\sigma^{\mu\nu}\gamma_5\Psi, gB_\mu J^\mu_{B-L}. \tag{2}$$

Here $H_{\mu\nu} = \partial_\mu B_\nu - \partial_\nu B_\mu$ is the gauge field strength of $B_\mu$. The first operator in (2) is a kinetic mixing between the dark gauge boson $B_\mu$ and the photon, the second and third operators are magnetic/electric dipole moments of standard model fermions under $B_\mu$ while the last is a vector current coupling between $B_\mu$ and a standard model current. In the last case, if we want $B_\mu$ to be anomaly free (so that it can be naturally light without invoking new degrees of freedom), the standard model current that naturally appears there is the B-L current (and its family dependent variants such as $L_i - L_j$ all of which lead to similar phenomenology). So we see that even though these bosons may span a large mass range, the demand of technical naturalness limits the possible interactions of these particles to 8 interactions.

A skeptical reader may ask if we should actually care about technical naturalness. After all, we now have very solid evidence of at least two fine tuned quantities in our universe - the cosmological constant and the higgs boson itself. Neither of these terms are protected by symmetry and the absence of symmetry did not prevent their existence, creating confounding theoretical problems. Our job as physicists is to figure out what is out there in the world instead of imposing philosophies on it - especially philosophies that are already empirically known to be violated. Indeed, it might be possible to solve naturalness problems via dynamical schemes or other manifestations of symmetry where the protection from radiative corrections is not immediately apparent. It is thus reasonable to consider a broader class of interactions than the ones described above where the interactions may naively yield radiative corrections to the boson's mass.

Interestingly, from the experimental point of view, we can ignore these theoretical arguments and ask a rather simple question: what are the possible ways in which a classical field can interact with standard model particles such as photons, electrons and nucleons - the objects that we can control and measure in the laboratory? There are only five possible effects:

1. The field can create photons - this effect can be caused by the operators $\frac{\phi}{f_\phi}F\tilde{F}$ and $\epsilon H_{\mu\nu}F^{\mu\nu}$.

2. The field can cause currents in circuits - this effect can also be caused by the operators $\frac{\phi}{f_\phi}F\tilde{F}$ and $\epsilon H_{\mu\nu}F^{\mu\nu}$.

3. The field can cause precession of electron and nucleon spins - this effect can be caused by the operators $\frac{\phi}{f_\phi}G\tilde{G}$, $\frac{\partial_\mu\phi}{f_\phi}\bar{\Psi}\gamma^\mu\gamma_5\Psi$, $\frac{H_{\mu\nu}}{\Lambda}\bar{\Psi}\sigma^{\mu\nu}\Psi$ and $\frac{H_{\mu\nu}}{\Lambda}\bar{\Psi}\sigma^{\mu\nu}\gamma_5\Psi$.

4. The field can exert forces on particles - this effect can be caused by the operators $g\phi h^2$ and $gB_\mu J^\mu_{B-L}$.

5. The field can change the values of fundamental constants - this effect can be caused by the operator $g\phi h^2$.

In the above cases, I have listed operators from (1) and (2) that yield these effects - this shows that these effects can arise from technically natural interactions. But, these effects can also be caused by technically unnatural interactions - for example, changes to fundamental

constants can be caused in dilaton interactions of the form $\frac{\phi}{\Lambda}F^2$ which effectively changes the fine structure constant. Thus, even if we decide to ignore naturalness, we are left with a rather small number of experimental signatures that we can probe.

How do these signatures look like? As discussed above, the dark matter is a time dependent classical field oscillating at a frequency equal to its mass - so we simply take this oscillating field and substitute it into the operators of interest (such as the technically natural interactions listed above - but this can also be done for any general operator). This then creates a time dependent term in the Lagrangian, giving rise to time dependent effects that we can observationally measure. Let us enumerate these effects and the experiments that can search for these effects in the following.

1. **Photon Production:** Take the operator $\frac{\phi}{f_\phi}F\tilde{F}$ - this is the canonical interaction of an axion-like-particle with electromagnetism. The field $\phi$ is now replaced by the classical dark matter field $\phi = \phi_0 \cos\left(m_\phi t - m_\phi v x\right)$ where $m_\phi^2 \phi_0^2 = \rho_{DM}$. With this substitution, we can derive the equations of motion of electromagnetism with the term $\frac{\phi_0 \cos\left(m_\phi t - m_\phi v x\right)}{f_\phi}F\tilde{F}$ in the Lagrangian. These are the equations of motion of axion electrodynamics - you will see that these equations allow for the conversion of a dark matter axion field into a photon in the presence of a background magnetic field. Moreover, there is a possibility of resonance since the dark matter field oscillates at a frequency $m_\phi$ with a width $\sim 10^{-6} m_\phi$. One can therefore imagine placing a resonant cavity in a background magnetic field and searching for the resonant conversion of the dark matter into photons. This is the Axion Dark Matter eXperiment (ADMX) [4].

   Similarly, the term $\epsilon H_{\mu\nu}F^{\mu\nu}$ also leads to the conversion of a dark matter vector boson into photons. To see this, observe that $H_{\mu\nu}$ is effectively a dark electric field and we can write it as $H_{\mu\nu} \sim m_B B_0 \cos\left(m_B t - m_B v x\right)$ where $m_B$ is the mass of the vector boson $B_\mu$ and its amplitude $B_0$ is determined by the local dark matter density $m_B^2 B_0^2 = \rho_{DM}$. Substituting for $H$ in the term $\epsilon H_{\mu\nu}F^{\mu\nu}$, we can derive the equations of motion of electromagnetism in the presence of this background dark matter field and we will see that it allows for the conversion of the dark matter into a photon. Once again, a resonance is possible since the dark matter field is a narrow band signal. Note that unlike the case of the axion discussed above, this conversion does not require a background magnetic field.

   Overall, we now have an idea for how to look for this effect. The dark matter field, under suitable conditions, excites modes of the photon that are at the same frequency as the dark matter. A suitable resonator such as an electromagnetic cavity would enhance this conversion. One can thus look for anomalous electromagnetic signals in a well shielded cavity (which is an electromagnetic resonator) and detect the dark matter. Now we don't know the mass of the dark matter - thus this resonator should be designed so that its resonant frequency can be continually changed. The experiment then looks for the dark matter at some frequency - if it does not find an interesting signal, the resonant frequency of the setup is changed and the experiment looks for dark matter at a different frequency. Thus, by scanning a whole band of frequencies, the experiment can look for a wide range of dark matter masses. One of the major experimental advantages of this kind of search is that the dark matter signal is narrow band and persistent - this makes it easier to combat a variety of experimental sources of background. If the experiment finds a signal at a particular frequency, it can simply sit there for a while and see if the signal is persistent - if it isn't, we know that we did not find the dark matter. For a persistent signal, the experiment can tune away to a different frequency and see if the signal disappears. If it doesn't, we know that we did not detect the narrow band

signature of dark matter. This fact distinguishes the search for oscillating ultra-light dark matter from conventional Weakly Interacting Massive Particle (WIMP) dark matter searches since the latter have to combat background over a wide range of frequencies as their signal is truly DC. In general combating DC sources of systematics is hard since they are both large (caused by a variety of human and natural sources) and are difficult to screen. But, if the dark matter was to be oscillating at a higher frequency, these noise sources are rapidly suppressed and more easily screened. Of course, this does mean that searching for low frequency dark matter (for e.g. $10^{-22}$ eV mass dark matter corresponds to a frequency of $yr^{-1}$) is more challenging due to the issues of combating DC noise.

While the above considerations were specifically discussed for the case of the conversion of dark matter into photons in a resonant cavity, the general strategy described above applies to all the ultra-light dark matter scenarios discussed below. In general, one thinks of designing a tunable resonant setup (where possible) and uses the ability to scan over the resonance to combat noise.

2. **Currents:** Whenever there is a physical process that can produce photons, that process can also be used to drive currents since electrons will respond to an electromagnetic field. The processes described above for the operators $\frac{\phi}{f_\phi} F\tilde{F}$ and $\epsilon H_{\mu\nu} F^{\mu\nu}$ will thus also drive currents. When is it useful to have the dark matter drive a current in a circuit as opposed to allowing it to resonantly convert in a cavity? An electromagnetic cavity is also fundamentally a circuit - so there isn't really a "deep" difference between the two concept. The key point though is compactness - in an electromagnetic cavity, the resonance frequency of the system is set by the physical size of the cavity. For dark matter with a mass greater than $\sim$ GHz, this corresponds to meter scale cavities. Cavities of this size can reasonably be built in the laboratory. However, if we want to find lower mass dark matter, we cannot quite use a resonant cavity since the physical size of the cavity will have to scale with the compton wavelength of the dark matter - this rapidly increases the complexity (and sheer real estate cost) of the experiment. To get around this difficulty, one needs a compact resonator *i.e.* a lumped element or LC resonant system that is able to achieve lower resonance frequencies without a similarly drastic increase in the physical size of the device (of course, it is not possible to make these frequencies arbitrarily small without a corresponding increase in the physical dimensions of the setup - but it is possible to gain a few orders of magnitude). This is the basic idea of the DM Radio (*i.e.* dark matter radio) experiment [7], where a LC resonator is placed instead a shield. The dark matter can resonantly excite this LC resonator creating a current in this circuit - the current creates a magnetic field which can be measured with a precise magnetometer like a Superconducting Quantum Interference Device (SQUID).

3. **Spin Precession:** Dark matter induced spin precession can be looked for using a variety of setups that are traditionally used in Nuclear Magnetic Resonance (NMR)/Electron Spin Resonance (ESR) experiments. If the dark matter induces nuclear spin precession, the basic idea is to take a sample of material and somehow polarize all the nuclei (this is far easier said than done - while nearly 100 % nuclear polarization has been experimentally demonstrated, it is far from being a routine task). The dark matter now causes the spin to precess, changing the magnetization of the sample. The change in the magnetization of the material can be measured using a precision magnetometer such as a SQUID. There is also a natural possibility of a resonance in this kind of system - one can turn on a background magnetic field which then sets the Larmor precession frequency of this system. A dark matter signal that is resonant with the Larmor precession frequency will give rise to an amplified response. Similar searches can also be performed to look for dark matter induced spin precession of electrons - here one would take a system

with polarized electrons and look for the change in the magnetization caused by the dark matter induced spin precession. Naively, it may seem advantageous to use electron spins over nuclear spins - first, electrons are a lot easier to polarize than nucleons and they have a much larger magnetic moment giving rise to larger signals in magnetometers for the same spin precession. However, a big factor in all these experiments is the amount of time for which the dark matter is able to drive the spin without the induced spin precession being damped by dissipative processes in the system. Since electrons interact strongly with each other, these dissipative processes are significantly stronger for electrons than they are for nuclei, significantly suppressing the signal in the electronic system. In the NMR/ESR literature, this is the so called transverse spin relaxation or $T_2$ time which is considerably larger for nuclei than it is for electrons. It is thus advantageous to search for nuclear spin precession as opposed to electron spin precession as long as the dark matter is able to cause both these effects at comparable levels. This is the basic idea of the Cosmic Axion Spin Precession Experiment (CASPEr) [5, 15].

It is useful to explicitly see how spin precession is induced by the various operators listed above. Begin by focusing on the defining coupling $\frac{\phi}{f_\phi} G\tilde{G}$ of the QCD axion. Substitute for $\phi$ as $\phi = \phi_0 \cos\left(m_\phi t - m_\phi v x\right)$ into this operator. Notice that this term has the same form as the infamous $\theta$ term of QCD which gives rise to the strong CP problem (this form is in fact the reason why the QCD axion can solve the strong CP problem). Since this has the same form as the $\theta$ term of QCD, non-perturbative QCD processes (instantons) will give rise to an electric dipole moment to nuclei that is proportional to $\phi_0 \cos\left(m_\phi t - m_\phi v x\right)$. Given the dark matter density, one can calculate that this is a very small dipole moment - many orders of magnitude smaller than the current limit on the static, time independent electric dipole moment of nucleons. However, the electric dipole moment induced by the axion dark matter is time dependent since it oscillates at a frequency $m_\phi$ - this time dependence can be leveraged to allow for new techniques that can be used to search for this effect. Naively, what we want to do is to polarize the nuclear spins in a sample and apply an electric field in a direction perpendicular to the spin polarization. If there is an electric dipole moment, much like how a magnetic dipole precesses under a magnetic field, an electric dipole will also precess due to an applied electric field. Thus, we see how the QCD axion can cause spin precession. To get the largest effect, we want a large number of spins and a large electric field. So what we want to do is to take a ferro-electric solid *i.e.* a solid whose unit cell lacks symmetry so that the central nucleus has a large, effectively atomic scale electric field (think of it as a polarized molecule, but in a solid). Now if the nuclear spin is placed in a direction that is perpendicular to that of this electric field, the spin will precess. A major advantage of the QCD axion induced effect is that this effect is naturally time varying and thus it can be read out without further ado. This is not the case for a static electric dipole moment since one would have to reverse the direction of the polarized unit cell to see this effect and such reversals give rise to new systematic effects (for example, from anomalous heating of the sample).

In fact the above description of the nuclear spin precessing due to its electric dipole moment being acted upon by an atomic scale electric field is "morally right" but in fact technically wrong. As pointed out by Schiff [23], the electric dipole moment of a nucleus that is in electrostatic equilibrium cannot be directly measured. This is because in electrostatic equilibrium, the net electric field at the location of the nucleus must vanish and thus there is no electric field for the nuclear dipole moment on the nucleus to couple to. The vanishing of the electric field occurs since the electron clouds in the system will move in such a manner so as to screen the electric field at the location of the nucleus.

Given this cancellation, how can we hope to see the nuclear electric dipole moment? We have to use the fact that the nucleus is not a point object and that while the electric field vanishes, its gradients need not. The gradients of the field will couple to higher order $T$ violating moments of the nucleus (also known as Schiff moments) which effectively give rise to an energy shift in the nucleus that depends upon the relative orientation between the electric field gradient (which is along the direction of the asymmetries of the ferroelectric unit cell) and the nuclear spin (which also sets the direction of the $T$ violating moments).

The spin precession caused by the other operators discussed above are more easily understood. In the non-relativistic limit, the operator $\frac{\partial_\mu \phi}{f_\phi} \bar{\Psi} \gamma^\mu \gamma_5 \Psi$ is of the form $\frac{m_\phi \phi_0}{f_\phi} \vec{v}.\vec{S}$ where $\vec{v}$ is the relative velocity between the spin $\vec{S}$ and the dark matter field. Thus, the relative velocity effectively acts like a pseudomagnetic field and if a spin is placed perpendicular to the direction of this velocity, it will precess. The other operators $\frac{H_{\mu\nu}}{\Lambda} \bar{\Psi} \sigma^{\mu\nu} \Psi$ and $\frac{H_{\mu\nu}}{\Lambda} \bar{\Psi} \sigma^{\mu\nu} \gamma_5 \Psi$ imply that standard model nucleons carry magnetic and electric dipole moments (respectively) under the new gauge boson. Thus, when a spin is placed perpendicular to the direction of this dark matter field, it causes the spin to precess.

4. **Accelerations:** The operators $g\phi h^2$ and $gB_\mu J^\mu_{B-L}$ directly induce accelerations on standard model particles in the presence of a background dark matter field. For the operator $g\phi h^2$ this arises from the fact that the dark matter field has a non-zero gradient $\nabla \phi \sim m_\phi \phi_0 \vec{v}$ in the galaxy resulting in a force exerted on particles that is in the direction of this gradient *i.e.* the direction of the relative velocity between the dark matter and the particle. For the operator $gB_\mu J^\mu_{B-L}$ the nature of the exerted force is immediate - the dark matter is basically a dark "electric" field and the standard model particle is charged under it, resulting in a force being applied to the particle from the dark matter field. This is a direct force exerted by the dark matter on the particle and not some kind of gravitational force (which of course exists - that is how we discovered the existence of dark matter). Since this is not a gravitational force, this force will violate the principle of equivalence *i.e.* the induced acceleration will depend upon the nature of the particle [14]. This is a useful way of thinking about these accelerations since this allows us to use a variety of setups that have already been constructed to search for equivalence principle violating forces (between matter, for example, the earth and a test body) to look for the equivalence principle violating force exerted by the dark matter on matter. The canonical example of an equivalence principle violating search are the torsion balance experiments of the Eot-Wash group. In these experiments, two different materials are placed at either end of a torsion balance - if there is an equivalence principle violating force (for example, between the earth and the materials), then the torsion balance will rotate and this rotation is precisely measured. Another possible setup are atom interferometry measurements that perform Galileo's famous experiment where two different isotopes are dropped and the relative acceleration between the isotopes is very precisely measured. One may basically use these existing setups and look for a time varying equivalence principle force - caused not by the earth exerting a new force on these particles but rather arising directly from the dark matter itself.

The time varying nature of the force is again helpful in combating systematic sources of noise that are confronted by these experiments. The major noise source that limits probes of static equivalence principle violating forces between the earth and test bodies is a gravity gradient *i.e.* since the two materials/isotopes will have a somewhat different location on the surface of the earth, they will fall differently due to the fact that the Earth's gravitational field is not uniform. This background is significantly suppressed

while searching for time varying equivalence principle violating forces since the accelerations induced by the dark matter are narrow band and can only be mimicked by time varying gravity gradient effects which are highly suppressed over large parts of the frequency spectrum.

In addition to equivalence principle violating experiments, these effects can also be searched for in gravitational wave detectors [14]. such as Pulsar Timing arrays and Laser Interferometer Gravitational wave Observatory (LIGO). In a pulsar timing array, the relative motion between a distant pulsar and a local base-station is measured. The dark matter will in general exert a very different force on the earth compared to the acceleration it exerts on a distant pulsar leading to a direct signal in these timing arrays. The one distinction between the dark matter signal and a gravitational wave signal in this experiment is the spatial morphology of the signal - unlike a gravitational wave signal that has a telltale quadrupole form, the dark matter signal would either be a scalar or a dipole signal. While there is the danger that this could get mixed with other solar system backgrounds, the dark matter signal would be narrow band and can thus likely be isolated. At LIGO, the signal in this setup arises from the fact that during the light travel time between the mirrors, the phase of the dark matter field changes resulting in a differential acceleration between the mirrors. This is not an equivalence principle violating effect but it is nevertheless a competitive measurement since LIGO is a terrific accelerometer with highly suppressed noise at its operating frequency.

5. **Fundamental Constants:** Classical fields can also cause time varying fundamental constants. For example, the operator $g\phi h^2$ results in a mixing between the dark matter $\phi$ and the higgs boson $h$. The higgs is responsible for giving mass to fundamental particles and thus when the dark matter oscillates, this results in a small oscillation of the masses of fundamental particles such as the electron. The electron mass sets the unit of energy in atoms and thus when the electron mass changes, the fundamental frequencies of atomic transitions are altered. This alteration manifests itself effectively as a relative acceleration between two well separated objects - this acceleration is not due to the fact that the objects are "really moving" away from each other - in fact, they are not. But, when the fundamental frequency changes, the unit of time/distance that is used to determine the distance between the objects changes, effectively appearing as a fluctuating distance between the two objects.

To see this effect, we may consider the thought experiment of sending photons at regular intervals between two well separated clocks. Suppose the first clock sends light pulses every $t$ seconds. If the second clock is at a distance $l$, these pulses will arrive at the second clock also spaced by intervals that are of size $t$. Now suppose there is a time variation in fundamental constants caused by the dark matter. This means that the frequencies of the clocks are continually changing as the electron mass continually keeps changing - so even if the first clock sends pulses at intervals of length $t$, the measurements of the arrival time at the second clock will fluctuate. Note that this effect depends upon the fact that in the light travel time between the two clocks, the phase of the dark matter field changes leading to a difference in the measured arrival time of the pulse relative to when it was emitted. In many ways, this is like a gravitational wave detector and thus gravitational wave detectors such as Pulsar Timing arrays and proposed single baseline gravitational wave detectors [3] can be used to search for this effect. This effect is also present in LIGO - but it is more difficult to see it in LIGO since the LIGO interferometer requires a differential signal along two orthogonal spatial directions in order to cancel noise from the laser. While such a differential signal exists for gravitational wave signals due to their quadrupole nature, such a differential signal gets cancelled in effects caused

by scalar dark matter since they are common to both directions.

We thus see that a large number of ultra-light dark matter candidates can be probed by focusing on the five main experimental effects they could have on standard model particles. While these effects are all present in technically natural theories, an experimentalist can simply focus on the nature of the experimental signature ignoring issues of technical naturalness. All of these searches are presently being performed, aided in part by the incredible advances that have occurred in quantum sensing enabling measurements of sub-femtotesla magnetic fields and accelerations at the level of $10^{-13}$ g or smaller.

## 3 Light Dark Matter

Existing WIMP direct detection techniques have successfully lowered their to thresholds to allow detection of dark matter with mass greater than $\sim 100$ MeV. We have just discussed many methods to probe ultra-light dark matter with mass between $10^{-22}$ eV - meV. It is however challenging to detect the absorption of bosonic dark matter in the mass range meV - eV and the elastic scattering of bosonic or fermionic dark matter in the mass range MeV - 100 MeV. In this mass range, the deposited energies are in the range meV - eV. But, these are not large enough to be visible in conventional experiments (although, there has been a continuous push to lower thresholds in many of these experiments). On the other hand, protocols to search for ultra-light dark matter that do not rely on the deposited energy leverage the coherence of the ultra-light dark matter signal to build a measurable phase in an experiment. The coherence of the dark matter signal is inversely proportional to its mass and at masses greater than $\sim 10^{-3}$ eV the coherence time is too small to employ phase accumulation techniques that we discussed above to detect ultra-light dark matter. How might one go about solving this problem?

One way to tackle this problem is to build some kind of amplifier *i.e.* when a small amount of energy is deposited in a material, we want an amplified response of the material so that we can easily observe that an interesting event has taken place. What kind of amplifier might we need to successfully probe dark matter? First, the amplification technology must be something we can utilize with a large target mass - otherwise, we are not going to be able to probe interesting dark matter cross-sections. Second, the amplification technology must be sufficiently stable over long periods of time *i.e.* we do not want the amplifier to go off by itself when no interesting events have occurred. Naively, this problem might not seem like it should be that difficult to overcome but in fact it is a serious limitation on many amplification technologies. For example, the simplest amplifier technology would be to apply a background electric field wherein even a low energy ionized electron would get accelerated as soon as it is ionized. However, the background electric field can cause events even when no energy is deposited in the detector - this is because random electrons stuck in various impurities can tunnel out of their local potentials and get accelerated by the background electric field. This phenomenon is known as the dark count rate and it is a common phenomenon in conventional photomultiplier tubes that amplify the effects of photon absorption. This problem becomes particularly acute when the threshold for the detector is lowered since the tunneling will occur more readily. Thus, in the parlance of amplifiers, what we want is a low threshold amplifier with a low dark count rate permitting us to operate the device for a long time.

The final ingredient that is necessary is the reset time of the device when an event occurs. We know that dark matter events are rare and that most events in the experiment are going to be background events from radioactivity. When we build an amplifier, it will amplify all energy depositions and thus radioactive backgrounds will also set off the amplifier. Once the amplifier is set off and we identify that the event is due to radioactivity, it will take some time for the amplifier to return to its ground state so that it is ready to see events again. This is the

reset time of the device - we want the reset time to be short compared to the expected rate of backgrounds in the detector so that we can have enough observation time between background events. The reset time is a crucial issue that blocks the use of conventional bubble chamber technology at low thresholds since the increase in background events at low thresholds results in frequent creation of bubbles and the time necessary to remove the bubbles and return the chamber to its ground state rapidly gets long, making it difficult to have long observation runs of the chamber.

How can we create a low threshold amplifier that can operate with a large target mass while simultaneously possessing a low dark count rate and a short reset time? One avenue [6] that could be pursued to tackle these problems is to investigate the use of single molecule magnets as "magnetic" bubble chambers. The basic idea is as follows. Suppose we take a system that has all its spins polarized in one direction. Let us now apply a magnetic field in the opposite direction of the spin polarization. Now, all of these spins are in a metastable state since they are anti-aligned with the external magnetic field. Suppose we deposit some small energy into this system which causes some of the spins in the region where the energy was deposited to relax to the ground state. When they relax to the ground state, the system will release the stored Zeeman energy in the system. This released Zeeman energy can diffuse into nearby regions causing those spins to also relax and thus release even more energy. This process thus sets off an avalanche of spin relaxation resulting in a macroscopic change to the overall magnetization of the system initiated by the microscopic process of energy deposition that relaxes some spins.

In what kind of material can we expect this above phenomenon to happen? It is difficult to realize this phenomenon in ferromagnetic materials where the spin-spin interaction is strong - in such a situation, causing any kind of spin flip will be energy intensive and thus it is difficult to create conditions where a small amount of initial heat can easily trigger spin relaxation. We thus need a system with weak spin-spin interactions. One way to create weak spin-spin interactions would be to look at lower density systems such as a gas. But in these cases, while the spin-spin interactions are weaker, the lower density makes it harder for heat to be conducted from the relaxed spins to their neighbors. We thus need a material where the spin-spin interaction is weak but the system is still at large density so that the heat from the relaxation can be efficiently conducted to the rest of the material. These kinds of conditions are satisfied in materials called single-molecule magnets - these are systems where the unit cell consists of a central complex that is magnetic (such as Manganese or Iron) surrounded by some number of organic elements. In such a system, the spin-spin coupling is weak since the distance between the magnetic complexes is a bit bigger due to all the organic material in the unit cell. At the same time, the organic material is able to efficiently conduct heat. In these systems, the spins in the unit cell effectively act as independent spins thus allowing the material to earn the name "single-molecule" magnets. These materials were investigated in the world of chemistry in the 1990s and early 2000s and the magnetic avalanche scenario described above has been experimentally witnessed in quite a few of these single-molecule magnet complexes. The chemists also know how to create large samples of these materials and thus it is possible, at least in principle, to obtain a large target mass (the chemists can produce kg scale powders of such materials, though the key issue for a dark matter experiment is radiopurity. The chemists do not care about radiopurity at the level necessary for a dark matter experiment and the ability to produce kg scale powders with the desired radiopurity has not been demonstrated.).

What is the basic physics of this phenomenon *i.e.* what are the conditions that determine the initiation of an avalanche? The key point is that in the absence of an applied external magnetic field, the spins in these materials can be described by a two level Hamiltonian where the two levels are exactly degenerate but with a potential barrier between them - think of this as something like a particle in a double well potential (see discussion in [6]). In the presence

of a magnetic field, this degeneracy is lifted with the creation of a metastable state (the state that is anti-aligned with the magnetic field) that is at higher energy than the ground state. This meta-stable state wants to decay to the ground state - but to do so it has to overcome the potential barrier. The lifetime $\tau$ of this system is described by the so-called Arrhenius law: $\tau = \tau_0 e^{U/T}$ where $\tau_0$ is an intrinsic time scale associated with the relaxation dynamics of the system, $U$ is the potential barrier between the states and $T$ the temperature of the system. From this equation, we see that when $T \gtrsim U$, the meta-stable state decays in the time $\tau_0$ while when $T \ll U$, the lifetime is very long. This is thus a system where at low temperature, we expect the system to be very stable (and thus potentially have a low dark count rate), but then a local change to the temperature can cause the meta-stable spins to rapidly decay leading to the release of stored energy. How large of an initial energy is necessary to trigger this avalanche? When heat is deposited into a local region in this system, there are two possibilities. Either the heat can cause the meta-stable spins to relax and initiating the avalanche or the heat can diffuse out of that region before the spins relax in which case the avalanche is not triggered. Now the relaxation time of the meta-stable state is controlled by the parameter $\tau_0$ - this parameter does not depend upon the physical size of the region that was initially heated to a certain temperature. The thermal diffusion time however cares about this physical size - heat diffusion proceeds as a random walk process and thus the time taken for heat to diffuse out of a region scales quadratically with the size of the region (we know this from the fact that small pieces of meat cook way faster than a large thanksgiving turkey). Thus, in any single-molecule magnet, if we heat up a sufficiently large volume of the material, the diffusion time scale will become longer than the relaxation time of the meta-stable state, creating the conditions necessary for an avalanche to exist. By using parameters such as the thermal condutivity of the material, the potential barrier $U$ and the relaxation time $\tau_0$, one can use the above condition to determine the threshold energy for the system. Using known properties of single-molecule magnets, there appear to be several examples of systems where the thresholds can be as low as 10 meV, making this an interesting direction to explore for low threshold dark matter detection. For most of these single-molecule magnets, the operating temperatures would be in the 1K - 4K range.

These materials also appear to have the ability to overcome the other main challenge of a low threshold amplifier - namely a short reset time. In these systems, with a fancy magnetometer like a SQUID, one does not need a large crystal to flip its magnetization in order to realize that an event of interest has occurred. With a sensitive magnetometer, it is sufficient for all the spins in a $\sim 100\,\mu$m region to flip and the magnetometer can read this signal in a $\sim 10$ cm sample within a few $\sim 10$s of $\mu$s. If we thus create a sample where the single-molecule magnets are grains of size $\sim 100\,\mu$m instead of one large crystal (this is in fact easier to do from the material science point of view - manufacturing large crystals is difficult!), whenever an avalanche goes off in the system, it will only relax the spins in one grain while the spins in the other grains will not be affected as heat will not be efficiently conducted between the grains. Effectively in this "solid bubble" chamber, the size of the "bubble" is automatically restricted to the grain size. Now if a background event occurs in this detector, it will cause relaxation in a $\sim (100\mu\text{m})^3$ volume while leaving the rest of the detector unaffected. There is thus no need to reset the detector after each background event. Depending upon the background rate (which has to do with the radiopurity of the sample), the detector will slowly lose operational volume over time - but this should still allow for significant observation time, based upon background rates that have been attained in more conventional dark matter experiments.

While there is a long road before this technology reaches the maturity to be used as a full scale dark matter experiment, there has recently been some experimental activity to demonstrate the key operational principles of this detector - namely, can a magnetic avalanche be triggered in an otherwise meta-stable single-molecule magnet crystal by the deposition of en-

ergy by particle scattering? This demonstration was successfully performed by a group at Texas A&M university where they showed that the energy deposited by $\sim$ MeV $\alpha$ particles can trigger avalanches in a meta-stable single molecule magnet crystal. This shows that single molecule magnet crystals can be used as particle detectors. Of course, from the dark matter point of view, this result by itself is not all that exciting - there is no need for fancy technology to detect $\sim$ MeV energy depositions. The reason for the $\sim$ MeV scale threshold in this experiment was set by the particular material chosen for the experiment whose relaxation time $\tau_0$ is quite long leading to a large threshold energy. It is hoped that a broader exploration of such materials with shorter values of $\tau_0$ will make it feasible to create lower threshold detectors.

# 4 Directional Detection

The above discussion has been focussed on detecting dark matter particles that have a mass much less than the weak scale. As you know, "weakly interacting massive particle" or "WIMP" dark matter has been extensively probed by a number of experimental teams for the past 30 years. The null results from these experiments are in fact a major motivation for looking at dark matter at other mass scales. In my view, while there is a considerable need to vastly expand the experimental program to look for a broad range of dark matter particles, continued probes of the WIMP are nevertheless well motivated and necessary. This is because in current direct detection experiments we are currently probing WIMP interactions with the standard model that are mediated by the higgs and there are a few more orders of magnitude in cross-section that we need to probe before we complete the probe of this higgs mediated scenario. Given the fact that the higgs exists and the fact that a weak scale particle with weak scale cross-sections can naturally be the dark matter (*i.e.* the WIMP miracle), it is important to advance this field forward. This field is however expected to hit a major background - the coherent scattering of neutrinos from the Sun. WIMP dark matter experiments utilize a variety of handles to reject a number of radioactive backgrounds, such as the fact that these radioactive backgrounds will typically scatter more than once in the detector, unlike the elastic scattering of dark matter. Unfortunately, the coherent elastic scattering of neutrinos from an atomic nucleus has the same event topology as dark matter scattering and the next generation of dark matter experiments are expected to be sensitive to solar neutrinos. If this background cannot be rejected, WIMP detection would require statistical discrimination of a small WIMP signal over a large background. This implies that the sensitivity of the detectors would only scale as $\sqrt{V}$ where $V$ is the volume of the detector. Since WIMP detectors are already at $V \sim m^3$, continued progress would rapidly require prohibitively large detectors.

One way to reject this background would be to identify the direction of the nuclear recoil induced by the collision of the dark matter (or neutrino). With such directional detection capability, one can make use of the fact that, due to momentum conservation, when a solar neutrino collides with a nucleus, the recoiling nucleus has to move away from the Sun. One could then reject all events that are pointed away from the known location of the Sun, eliminating the neutrino background. Incident WIMPs are expected to be isotropic; and thus by focusing only on events where the recoil is not along the direction of the Sun, one will be able to only look at events caused by dark matter. Such a directional detector will suffer a loss of sensitivity of $\sim$ 50 percent while dramatically reducing the neutrino background. In addition to overcoming the neutrino background, such a directional detector could potentially also be used to detect the direction of the dark matter wind. It is thus of great interest to develop techniques to measure the direction of the nuclear recoil induced by a dark matter/neutrino collision.

The technical problem that must be overcome for directional detection is the following.

The scattering of dark matter/neutrino deposits energies $\sim$ 10 - 30 keV. The direction of the induced nuclear recoil must be established in a detector with a large target mass, to overcome the tiny WIMP/neutrino cross-sections. To accommodate the large target mass without having to resort to enormous detector volumes, it is advantageous for the detector to be a high density material like a solid or a liquid. While there are excellent directional detection techniques in gas-based detectors, there are no well established techniques for directional detection in high density materials.

What kind of signature can we look for in a high density material that is sensitive to the direction of the nuclear recoil? Interestingly, there is an observable signature in a solid state system that is sensitive to this direction. When a dark matter/neutrino collides with a nucleus and deposits $\sim$ 10 keV of energy, this nucleus is kicked out of its location in the lattice since the lattice potentials of solids are $\sim$ 10 - 20 eV. This nucleus moves through the lattice and scatters with other nearby atoms, knocking those atoms off their locations in the lattice as well [22]. Much like a bullet going through a solid and leaving a damage trail that is correlated with the direction of the bullet, the recoiling nucleus creates a bunch of damage in the lattice with the damage cluster being correlated with the initial direction of the nuclear recoil. It can be shown via simulations that for a $\sim$ 10 keV nucleus, there is a tell-tale damage cluster that is about $\sim$ 100 nm in size which is well correlated with the direction of the nuclear recoil. In this damage cluster, the number of lattice vacancies and dislocations are $\sim$ 100 - 200, considerably larger than the number of crystal defects that one would get in a pure crystal at this length scale. There is thus a robust signature of the nuclear recoil in a solid. The difficulty is that this signal is localized to within $\sim$ 100 nm and thus one needs to find this signal in a large target volume - the proverbial problem of finding a needle in a haystack.

To tackle this problem, we should first ask if it is possible to perform nanoscale sensing in a solid. Fortunately, the answer to this question is yes (see references in [22]). One can perform spectroscopy of crystal defects in a solid. The basic idea is to consider crystals which possess point quantum defects *i.e.* we take a nice well ordered crystal and replace some elements of that crystal with another element. An example of such a system is diamond with nitrogen vacancy centers where some of the carbons in a diamond lattice are replaced by nitrogen and this nitrogen co-exists with another carbon vacancy. The energy levels of the electrons in these defects is sensitive to the local electronic environment. By performing spectroscopy of these electrons (for example with a laser) we can determine this local electronic state. Crystal damage induced by the recoiling nucleus will cause strain in the crystal and this strain will shift these electronic levels, making it possible for the damage to be measured via spectroscopy. This sort of spectroscopic measurement of crystal damage can likely be done in a variety of materials with point quantum defects (such as F centers of metal halides) - but for the purpose of this discussion we will focus on the possibilities of diamond with nitrogen vacancy centers. This is simply because these nitrogen vacancy centers are well studied and many of their properties and capabilities have been demonstrated in the laboratory. In these, one can show that the line shifts induced from the strain caused by $\sim$ 10 keV nuclear recoils are $\sim$ 30 kHz, considerably larger than the $\sim$ 300 Hz linewidths of the nitrogen vacancy center itself. Thus, locally, we see that the signal to noise ratio is $\sim$ 100 and as long as we are able to localize the volume where the damage occurred, we can reasonably hope to be able to read the signal out.

How can we localize the volume where the damage occurred? The key point to note here is that we are interested in perhaps $\sim \mathcal{O}(10 - 100)$ events of interest in a large target mass that could be from WIMPs/neutrinos. In an experiment with an operating time $\sim$ year, we have $\sim$ a day to study an event of interest to determine its direction. The following protocol could be adopted to achieve this goal. We will imagine taking a sectioned detector where each section has thickness $\sim$ mm - but the lateral area of this section can be large (potentially $\sim m^2$). Several of these sections are stacked on top of each other to create a large target volume.

Now suppose an event occurs in this detector - typically from background and rarely from a neutrino/dark matter. One uses standard WIMP detection techniques such as the identification of electron vs nuclear recoils and vetoing of multiple scattering events to focus on the small ($\sim \mathcal{O}(10-100)$) number of events that could be from dark matter or neutrinos. For these events of interest, conventional technology (such as the collection of scintillation light) can be used to localize the positions of these events to within a volume $\sim$ mm$^3$. Since our detector is sectioned to $\sim$ mm in thickness, we now know which section of the detector contained this event of interest. The technical problem now reduces to identifying the damage trail within this $\sim$ mm$^3$ volume. Since the crystal damage is stable, we can pull out this section of the detector and study it for $\sim$ a day to map out this damage trail. This mapping out is done in two stages. First, by simply shining laser light and looking for the fact that the light would not be absorbed by the damaged part of the crystal, we can localize the damaged region to within a wavelength of the light $\sim \mu$m. We now need to perform sub-wavelength resolution to further constrain the location of the damage trail. This can be done by applying an external magnetic field gradient $\sim$ Tesla/cm (this explains the need for the sectioned detector) - under this magnetic field gradient the nitrogen vacancy center lines shift in frequency in a position dependent way and by shining light of various frequencies into this system, sub-wavelength resolution can be obtained.

Many of the elements of this protocol have been independently demonstrated in the laboratory. For example, spectroscopy of nitrogen vacancy centers has been performed with nanoscale resolution. The damage trails created by nuclear recoils have been imaged to be at the $\sim 100$ nm scale in emulsion films (which is another possible way to detect the direction of nuclear recoils). We thus know that individual elements of this physics program make sense. It remains to be seen if they can be integrated in the laboratory into one single package. In addition to the spectroscopy of nitrogen vacancy centers, it is also likely that other methods to detect crystal damage can also be used to identify these tracks.

## 5 Ultra-Heavy Dark Matter

All current dark matter detection strategies, ranging from direct detection efforts in the laboratory to indirect signals from the annihilation (or decay) of dark matter, are based on the assumption that the dark matter is distributed around the universe as a gas of free particles with a reasonably large number density.[2] This large number density yields a high enough flux of dark matter enabling the detection of rare dark matter events. This picture of dark matter as a gas of free particles naturally emerges if self interactions within the dark sector are weak. What if the dark sector had strong self interactions?

In this case, much like the standard model undergoing nucleosynthesis and producing composite nuclei, the dark sector will also undergo a nucleosynthesis process in the early universe that may be highly efficient since it need not suffer from the accidents of nuclear physics in the standard model that inhibit the production of heavy elements. As a result, individual dark matter particles could coalesce to form very large composite states. Observational constraints on these self-interactions are weak. The most stringent constraints arise from observations of the Bullet Cluster, restricting these self interaction cross-sections to be less than approximately 1 cm$^2$/g. Since this bound is based on the dark matter distribution today, it is significantly weakened if the dark matter is clustered into heavy composite states with a low number density.

Given that the standard model, despite the peculiarities of nuclear physics, produces a huge range of composite states, it is highly likely that a variety of composite objects are likely

---

[2]The exception are searches for dark matter with astrophysical scale mass, such as Primordial Black Holes.

possible in complex dark sectors (for *e.g.* see [8]). If this is the case, the flux of the dark matter through any detector would be very small due to the small number density of these large agglomerations of dark matter. What are generic strategies that we may employ to look for such dark matter? Any such strategy must confront the fact that the number density of these events is low. This implies that the strategy must be extendable to large space-time volume detection. In doing so, the strategy must leverage the fact that the transit of a large composite dark matter state will likely be far more spectacular than single WIMP scattering events since the composite state carries a large number of dark matter particles potentially allowing for many possible ways for the standard model to interact with the dark matter during any of these transits. We focus on two distinct cases. In the first, discussed in sub-section 5.1, we focus on dark matter states that can be probed using terrestrial detectors. In sub-section 5.2, we focus on dark matter states that can trigger dramatic nuclear instabilities in white dwarfs, enabling us to probe a very different kind of dark matter.

## 5.1 Terrestrial

In the first part of this section, we will be interested in experimental strategies that can be adopted to look for the active transit of a composite dark matter object through the detector *i.e.* the transit occurs while the detector is actively monitoring the detection volume. In the second part, we will look at paleo detection, a concept where we look at tracks left by the dark matter in a transit that occurred a long time before the detector was constructed (or for that matter, conceived).

### 5.1.1 Active Detection

As pointed out earlier, the transit of these kinds of dark matter is a rare event due to its number density but these rare transits have the ability to cause observable effects in detectors due to the large number of particles in the dark matter object. A search for these rare events requires methods to distinguish it from backgrounds. There are two potential handles that could be exploited to achieve this goal. First, the dark matter moves with a speed $\sim 220$ km/s, significantly faster than any terrestrial source of noise, but significantly slower than the speed of light, placing it in a unique range of speed between terrestrial and cosmic ray induced events. If the signal from the dark matter is large enough to be observed at multiple locations in a detector that also has sufficient temporal resolution, it should be possible to distinguish this signal from other background transients. These events should also lie along a straight line, enabling further background rejection. Second, the dark matter has the ability to pierce through shields and interact in its own unique way with standard model sensors. Thus, in a setup that is monitored with a variety of precision sensors, the collective information from all sensors could potentially be used to reject standard model backgrounds. This latter option is technically challenging, but it is similar in spirit to WIMP detection experiments that use data from multiple channels to veto standard model events. Similar protocols could also be employed in experiments such as LIGO which monitor a variety of potential noise sources.

With these comments out of the way, let us write down a simple model for these composite dark matter states. The composite dark matter is constituted from partons which I will label $\chi$. The $\chi$ could be fermionic or bosonic. In the fermionic case, due to Fermi degeneracy, the physical radius of the composite object will increase as $N_\chi^{\frac{1}{3}}$ where $N_\chi$ is the number of partons in the object. For a bosonic theory, the physical radius of the composite state is model dependent - since there is no fundamental reason why a large number of bosons cannot be packed into the same quantum state, the physical size of the object will depend upon the details of the interactions between the bosons. My objective here is to identify qualitatively new signatures of these composite states and thus I will not investigate the details of any

particular model. But this does not mean that the details of the model are irrelevant - on the contrary, while mapping the discovery reach of any particular experiment to the underlying model parameters, these details are extremely important - for example, a compact bosonic object can more easily engage in enhanced coherent scattering that transfers high momentum rather than a more diffusively spread out fermionic object. Due to the enhanced cross-section, the parameter space of bosonic objects that can be probed is significantly bigger than the corresponding parameter space for fermionic objects. While the parameter space that can be experimentally probed is model dependent, the broad class of observational signatures that can be probed is not particularly sensitive to whether the partons are fermions or bosons.

What kinds of observational signatures can we go after [12] ? We consider the following toy model where we couple the parton $\chi$ to a bosonic mediator $\phi$ via a yukawa interaction and allow $\phi$ to interact with the standard model. The lagrangian is:

$$\mathcal{L} \supset g_\chi \phi \chi \chi + \mu^2 \phi^2 + g_N \phi \bar{N} N + \frac{\partial_\mu \phi}{f} \bar{N} \gamma^\mu \gamma_5 N + \frac{\phi}{\alpha M} \phi F_{\mu\nu} F^{\mu\nu}, \tag{3}$$

where $\mu$ is the mass of the mediator. In the above, I have been somewhat loose about $\chi$ - if $\chi$ is a boson, the interaction strength $g_\chi$ has dimensions of mass while if it is a fermion it is dimensionless (and thus more properly a yukawa coupling). For a fermion the appropriate operator would be written as $g_\chi \phi \bar{\chi} \chi$ while for a boson it would be $g_\chi \phi \chi^* \chi$ instead of the loose form I have described it above. $N$ refers to nucleons and $F_{\mu\nu}$ the field strength of electromagnetism.

The observational signatures of this scenario depend a lot on the mass of the mediator $\mu$. If we have a standard model probe of this composite state by a nucleus of mass $m_N$, when the mediator mass $\mu$ is such that $\mu \lessapprox m_N v$ where $v \sim 10^{-3}$ is the relative velocity between the dark matter and the nucleus, the mediator can basically be viewed as a long range interaction while when $\mu \gtrapprox m_N v$ the interaction is short ranged. For short ranged interactions, the observational signatures are due to scattering and energy deposition and this will be the first set of signatures that I describe. Following this, I will talk about the signatures that one can look for in the long range case.

For short ranged interactions, there is an aspect of the underlying model that greatly impacts the observational signatures of these scatterings. This is the energy scale $\Lambda$ associated with the "Bohr radius" of the parton in the composite state. In these scattering interactions, we are asking for the cross-section for a nucleus to scatter off the composite object. At the partonic level, the scattering occurs between the nucleus and the parton - the parton then has to transfer the momentum it gained in the collision to the rest of the composite object. How large of a momentum can be exchanged in this process? The "Bohr Radius" of the parton in the composite state $\sim 1/\Lambda$ sets the scale of the momentum uncertainty in the parton and it can be shown that for momentum exchange $\gg \Lambda$ the scattering cross-section is form factor suppressed (this is familiar from the corresponding phenomena in standard WIMP dark matter collisions where one computes a nuclear form factor). Thus, the scale $\Lambda$, which is a free parameter in this story, determines the largest momentum that can be exchanged in these interactions. When $\Lambda \lessapprox 300$ keV, the energy transferred to the nucleus is too small to cause the ionization in the detector. Thus for low $\Lambda$ the energy will be deposited in the form of heat as opposed to ionization. For $\Lambda \gtrapprox 300$ keV, enough momentum can be transferred in the collision to cause ionization. The other parameter that sets the largest momentum that can be exchanged in these collisions is the reduced mass of the composite dark matter and the nucleus - since the composite dark matter we are interested in is extremely massive, the reduced mass is simply set by the mass of the nucleus $m_N$. Thus, the largest momentum that can be exchanged in these collisions is the smaller of the scale $\Lambda$ and the nuclear de-Broglie momentum $m_N v$.

The main observational signature that one can look for in this case is the fact that dark matter can cause multiple scattering in the detector - these scatterings may be in the form of

ionization or simply the deposition of heat depending upon the unknown scale $\Lambda$. We can aim to distinguish these events from backgrounds by using timing information - if the events are due to dark matter, the scattering will be temporally separated in a manner consistent with the $\sim 10^{-3}$ velocity of dark matter in the galaxy. Searches for these kinds of scattering can be implemented in existing experiments such as Xenon that have large volume and are sensitive to ionization or in setups such as CDMS/EDELWEISS/CRESST that have excellent calorimetry and thus are sensitive to depositions of heat without corresponding ionization signatures. It can be shown [12] that there are robust models that can be tested in these experiments that are consistent with all other observational bounds on these scenarios.

In the case of long ranged interactions, the composite dark matter state sources a classical $\phi$ field outside it. It can now interact with standard model particles via any of the interactions described in (3). What are the observational effects of these interactions? These interactions are of the same form as the ones we described in section 2 - namely, these are interactions of a classical field with the standard model. Just as in that case, these classical fields from the dark matter will cause the effects discussed in section 2 - namely, it can produce electromagnetic waves, cause currents, induce spin precession, exert forces on bodies and change the values of fundamental constants. The key difference between this scenario and that of the ultra-light dark matter is that in the latter case the signals are persistent and occur as a narrow band signal around the dark matter mass. In this case, the signals are transient - they exist only when the composite dark matter transits near the detector and thus we need a different way of searching for them. Specifically, we don't need the resonant detection strategies that are adopted to look for ultra-light dark matter - rather we need broadband devices that are sensitive to these classical transits. Now in any one device, these transits will look like some sudden source of noise and we will not be able to distinguish it from backgrounds. But if we built a network of such detectors, we can do cross-correlations that search for effects that are correlated with the $\sim 10^{-3}$ velocity of the dark matter through the earth and this would allow this network to be uniquely sensitive to these kinds of dark matter objects. The GNOME project is an example of this kind of detector.

### 5.1.2 Paleo Detection

Another attractive avenue that could be pursued to overcome the low flux of extremely heavy dark matter is the concept of paleo detection. In these, the main idea is to look for the distinctive tracks left by the transit of dark matter in an old piece of rock [10] - the long temporal exposure of the rock, which could potentially be as old as the earth itself, gives this kind of search access to a large space-time volume, enabling it to combat the low flux of extremely heavy dark matter. It is also natural that extremely heavy dark matter can cause tracks that are distinctive from other sources of tracks in the rock such as geology and radioactivity. The key point is that the energy scales of dark matter are completely different from the energy scales $\lesssim 10$ eV associated with chemical or geological processes. Thus the dark matter transits can easily mess with the lattice structure of the rock. Second, extremely heavy dark matter can plow through the rock leaving a very long damage trail - this distinguishes it from the short stumps that are caused by radioactivity. Moreover, these tracks will also be continuous unlike the ones caused by cosmic rays. In the case of cosmic rays, particles such as muons lose energy continuously through electromagnetism - but these energy depositions do not cause significant lattice damage while collisions from protons and such do not cause continuous lattice damage. The way these experiments would operate is to take some area of a rock and scan its surface to find interesting defects - if such a defect is found, the region around that defect can be probed to see if the defect continues to exist as expected from a dark matter event. The sensitivity of the detector is limited by the scanning time required to perform the initial scan of the area to identify locations of interest.

Paleo detection is in fact a well established concept. Pioneering measurements were performed by Price and Salamon [**?**] where they looked for evidence of particle tracks in ancient mica. The main limitation of their method was that they needed very pure samples of mica - this was tied to their readout scheme. In their readout scheme, to discover the tracks caused by the transit of dark matter, they used the process of acid etching. This process enlarges the size of the tracks caused by a potential particle transit so that it can be observed by an optical microscope, but it also enlarges other defects in the sample. Thus if the sample is not particularly defect free, their method will be limited by this background.

More modern methods of imaging these tracks can potentially get around these problems - for example, a scanning electron microscope (SEM) can be combined with a cathodoluminescence (CL) detector which allows imaging of rock samples without acid etching. This technique [10] allows for the use of less pure rock samples, potentially permitting the use of several $\sim m^2$ of rock to be analyzed. In the initial versions of this proposed experiment, the plan is to search for tracks with radius $\sim \mu$m, which is large enough to be efficiently analyzed via SEM-CL detectors. Tracks of this radius are assumed to be caused by the transit of dark matter which deposits enough heat in the rock to cause melting along this $\sim \mu$m radius cylinder. Thus the signal would be something like a very long $\mu$m radius melted defect in the rock with none of the other regions surrounding this defect being deformed in any way. It is difficult to produce such a well localized defect via geological processes.

## 5.2   White Dwarfs

Similar to the concept of paleo detection articulated above, it is interesting to ask if the energy deposited by a rare dark matter event could cause a truly spectacular event in an astrophysical object that could be easily witnessed from the earth. Interestingly, the answer turns out to be yes. White dwarfs can be made to explode via localized deposition of energy [16]. When they explode, they will blow up as a Type 1a supernova. These explosions can happen even when the mass of the white dwarf is well below the Chandrasekhar limit[3]. Thus, one can use the existence of sub-Chandrasekhar white dwarfs or the observed Type 1a supernova rate to put constraints on such localized energy depositions from dark matter. Alternately, since the exact mechanism that causes certain Type 1a explosions is currently unknown, there is the intriguing possibility that they may be caused by dark matter events.

Before we discuss the physics of these explosions, let us observe that using white dwarfs as dark matter detectors allows for a detector of unique capability. First, the white dwarfs are old. Second, their ability to capture dark matter extends will beyond their $\sim 10^4$ km size - they can gobble up dark matter from even outside their physical size due to enhanced gravitational capture arising from the fact that the escape velocity of the white dwarf $\sim 10^{-2}$ is larger than the virial velocity $\sim 10^{-3}$ of the dark matter. This is thus a truly gigantic dark matter detector - it spans an enormous space-time volume! Finally, when the white dwarf explodes, the explosion is visible all over the universe enabling us to build a robust detection capability.

Why are white dwarfs susceptible to this kind of phenomena? The key point is that the white dwarf contains nuclei that would like to fuse - these are nuclei like Carbon/Oxygen or Oxygen/Neon/Magnesium. In a typical white dwarf, the temperature of the white dwarf is $\sim$ keV - at these temperatures, the Coulomb barrier prevents the fusion of these nuclei. But, the nuclear fusion rate is a very strong function of temperature - as the temperature increases, the fusion rate becomes exponentially larger and the reactions can occur without any suppression. Second, the white dwarf is supported by electron degeneracy pressure - this has the extremely important consequence that the density and pressure in the star are quite insensitive to the temperature. When the temperature in a part of the star increases, the density and pressure

---

[3]In the standard model, white dwarfs with a mass below the Chandrasekhar limit are expected to be stable.

do not change - this is unlike a star like the Sun which is modelled as an ideal gas. In the latter case, when the temperature of a local region is increased, the region expands in size and cools down - stars supported by an ideal gas equation of state are thus stable in the sense that they have a temperature regulation mechanism. This is not the case for the white dwarf - when temperature increases, it does not automatically decrease due to expansion. These two facts make the white dwarf susceptible to runaway fusion.

Suppose you take a small part of the white dwarf and you locally heat its temperature to $\sim$ MeV which is $\gg$ keV, the ambient temperature of the star. This increased temperature can do one of two things. With the disappearance of the Coulomb barrier, this can cause fusion. Or since the temperature does not automatically decrease by coherent expansion, it can diffuse out of the region. If the fusion rate is faster than the diffusion rate, similar to our discussion of the magnetic bubble chambers in section 3, the nuclei in this region will all undergo fusion releasing $\sim$ MeV of energy per fusion and this released energy will cause other nuclei to also fuse, triggering a nuclear explosion of the white dwarf as a Type 1a supernova. This process can occur well below the white dwarf reaches the Chandrasekhar limit. Now, the minimal energy necessary to achieve this fusion process is set by the condition that the fusion rate should be larger than the thermal diffusion rate. Since the latter is set by a random walk process, it cares about the size of the initial region that was heated up - but the fusion rate does not care about this size. Thus, given any white dwarf, there is a minimal size that needs to be heated up to a certain temperature and once that has been achieved, the fusion rate will dominate the thermal diffusion rate and the system will explode as a Type 1a supernova.

The minimal energy needed to cause such explosions have been calculated carefully by the Type 1a supernova community [25]. The exact energy necessary for the explosion depends upon the density of the white dwarf which turns out to be a strong function of the mass of the white dwarf. Second, it also depends upon the composition of the white dwarf - Carbon/Oxygen white dwarfs need less energy than Oxygen/Neon white dwarfs. While these details are important to analyze specific dark matter models, broadly speaking, these energy depositions range from $\sim 10^{16}$ GeV - $10^{21}$ GeV for a wide range of scenarios. The key point in these energy depositions is that the deposited energy must be localized - typically within $\sim 10^{-5} - 10^{-3}$ cm in order for the local temperature to be high enough to trigger such explosions.

As a simple application of these bounds, one may place limits on the transit of primordial black holes [16]. As the black hole transits, it will, via dynamical friction cause nuclei that are close to its trajectory to get hot. Dynamical friction is simply the process by which nuclei on either side of the black hole trajectory get accelerated to a high velocity and causes them to hit each with a higher velocity *i.e.* gain temperature. One thus gets a narrow hot ($\sim$ MeV) cylinder that is along the trajectory of the black hole as it moves through the star. If this cylinder is sufficiently thick, it will cause the white dwarf to explode as a Type 1a supernova, even though its mass is below the Chandrasekhar limit. The thickness of this cylinder is set by the black hole mass - the larger the mass of the black hole, the larger is the region that gets hot due to gravitational attraction, making it possible to blow up white dwarfs more easily. One may use the existence of populations of sub-Chandrasekhar white dwarfs and the observed Type 1a supernova rate to place limits on primordial black hole dark matter in the mass range $\sim 10^{19}$ gm - $10^{24}$ gm. Similar bounds can also be placed on a variety of dark matter particles [13, 19] as well as cosmological stable charged relics [11]. While this scenario currently appears to be a way to limit the properties of dark matter, given the absence of convincing astrophysical explanations for a variety of observed sub-Chandrasekhar supernovae [20] (especially the highly unusual Calcium-rich transients) it would be interesting to think of observational ways to probe if some of these supernovae could in fact be triggered by dark matter.

# 6 Conclusions

The identification of the nature of dark matter is pretty clearly one of the major problems confronting particle physics. It is exceedingly unlikely that humanity will solve this problem from the armchair by guessing a sufficiently pretty theory. Physics is an experimental field - the belief that we can figure out what is out there in the world without experimental input has always just been a silly fantasy. Given the vastness of the parameter space of dark matter, there is a tremendous need to dramatically widen the experimental program that has been pursued to detect its properties. Now, it could have been the case that this dramatic widening could only come at great cost - if every probe of a part of dark matter parameter space required billions of dollars and thousands of working hours, we will not be able to appreciably probe the dark matter parameter space in our lifetimes. Luckily, this is not the case - the methods and experiments described in these lectures are experiments that can be pursued by a small number of investigators at the cost of several million dollars per experiment. It is thus possible to sustain a robust ecosystem of dark matter experiments which will cover a significant range of parameter space. While the creation of such a program is not up to me, I certainly hope that this broad ranged program will come to be realized.

## Acknowledgments

I am grateful to Michaelangelo Traina for feedback on earlier drafts of these notes. S.R. is supported in part by the U.S. National Science Foundation (NSF) under Grant No. PHY-1818899. This work was supported by the U.S. Department of Energy (DOE), Office of Science, National Quantum Information Science Research Centers, Superconducting Quantum Materials and Systems Center (SQMS) under contract No. DE-AC02-07CH11359. S.R. is also supported by the DOE under a QuantISED grant for MAGIS, and the Simons Investigator Award No. 827042.

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
