# Peer review of "New Directions in the Search for Dark Matter"

_SciPost Physics Lecture Notes, doi:SciPost Phys. Lect. Notes 56 (2022)_

## Round 1 · Referee Report · Anonymous (Referee 3) · 2021-11-24

Strengths
1- Easy and enjoyable reading
2- Well described interplay of theoretical predictions and experimental observations
3- Clear description of what is and what isn't to be expected from these lecture notes
4- Motivating insight into new strategies while highlighting the importance of a rich and diverse experimental field
Weaknesses
1- Abbreviating terms are rarely introduced - this can reduce the level of understanding for people who don't know what they mean
2- Not many references are given; more references would allow the reader to learn more about, or remind her-/himself of, some aspects/terms that are only briefly dropped (the strong CP problem, the hierarchy problem, relaxion models, LIGO and almost all other mentioned experiments, Chandrasekhar limit...) - as it is the manuscript is focusing more on people who are already expert to some extend as opposed to being accessible to a wider range of scientists and especially students
3- In some sections the text is too qualitative to get an order of magnitude idea of things like devices and detectors. A few more, rough numbers here and there would be useful.
Report
I recommend the publication of these lecture notes in SciPost after a few mostly minor comments have been addressed. Overall this is really pleasant reading and the notes are shining light also on some less traditional approaches in a motivating and encouraging way - just the right reading for a new generation of scientists.
Requested changes
1-Eq. (1-3): Please introduce/name all of the individual terms that haven't been introduced before. I agree, some of which should be known. But for completeness and as example for good scientific writing I'd appreciate the addition of them.
2- I'd kindly ask the author to add a few more references throughout the manuscript based on his own judgement (see my comment in the Weaknesses section).
3- Sec. 3 states at the beginning "Existing WIMP direct detection techniques have successfully lowered their thresholds to allow detection of dark matter with mass greater than 100 MeV. We have just discussed many methods to probe ultra-light dark matter with mass between 10^-22 eV - meV. It is however challenging to detect matter in the mass range meV - 100 MeV. In this mass range, dark matter can deposit energies meV - 10 eV through absorption and inelastic scattering. But, these are not large enough to be visible in conventional bolometric experiments.".
I have a few comments/questions here which I hope the author can address to make this paragraph more clear and more accurate:
3.1- Why is the last sentence only referring to bolometric experiments? That sounds very selective for no obvious reason. Also, thresholds down to ~1eV depositions of energy are achieved by now (in bolometric and non-bolometric direct DM experiments, though, granted, one could argue about how conventional they are)
3.2- It is unclear what kind of DM is referred to, i.e. is bosonic DM also considered at this point? If so, then the absorption of meV-100MeV range bosonic DM (for which viable CDM candidates exist) can lead to deposited energies in the same range, i.e. meV-100MeV. And at least deposited energies >~1eV are observable nowadays with direct DM detection techniques including bolometric ones.
3.3- For the same reason (i.e. the question of the type of DM considered), it is unclear whether an achieved DM mass threshold of 100MeV (first sentence) should be quoted or lower. Thermally produced DM can nowadays be observed down to a mass of ~1MeV. Bosonic DM can be observed down to a mass of ~1eV.
4- Sec. 3: I'm lacking a few quantitative examples. This section very nicely(!) describes single molecule magnets. I'm not an expert on this particular subject and would be curious about the rough temperature at which a typical smm has to be operated. Maybe such an example number can be given at the end of the paragraph ending on p. 16.
Furthermore, p. 15 says "The chemists also know to create large samples of these materials and thus it is possible, at least in principle, to obtain a large target mass". How large is "large" roughly? Target masses in the direct DM community range from gram-scale to kt-scale. So large to me in this context means the latter but certainly for smm's we're closer to the g-scale (which I'd in fact call small). Please provide some numbers here (similar to how it is done in Sec. 4).
5- p.23: "It can be shown that there are robust models that can be tested in these experiments that are consistent with all other observational bounds on these scenarios." Is there a reference that can be added? If so, please do so.
6- Sec. 5.2 title: change "White Dwarves" to "White Dwarfs" (in agreement with the rest of the text).
7- Sec. 6: Supporting the role of women in science I suggest to change "thousands of man hours" into "thousands of working hours".

---

## Round 1 · Referee Report · Anonymous (Referee 2) · 2021-12-4

Report
These lecture notes remarkably include just three equations, and yet they aim at covering experimental searches of dark matter across 79 orders of magnitude. Although they
should probably be complemented with more pedagogical material, they flow very nicely and give a glimpse of the new exciting developments in the field.
I would hence recommend for publication in the SciPost Physics Lecture Notes section, after some small corrections are implemented (see below).
Requested changes
1. Around eq. (2) it is argued that the B-L is the only SM current associated to an anomaly-free gauge boson (so that it can be naturally light without invoking new degrees of freedom). I understand that just an example would suffice here, but the statements above are not correct. To be precise there are other family-dependent currents like L_i - L_j that are anomaly-free in the SM and they do not even require new degrees of freedom to be anomaly-free (differently from B-L which would require RH neutrinos to cancel anomalies).
2. Some acronyms might not be obvious to an (astro)particle reader: NMR/ESR, SQUID, DC, … Here, I am imagining that the main target of these lecture notes are (astro)particle physicists. Hence, it would be helpful not to give for granted many condensed matter concepts.
3. In sect. 5 it would be useful a more accurate description on how ultra-heavy dark matter could be produced without overshooting dark matter relic density.

---

## Round 1 · Referee Report · Anonymous (Referee 1) · 2022-1-23

Strengths
1- Inspirational text on a variety of techniques in Dark Matter detection. Exactly in scope for lecture notes at a summer school on the topic
2- Clarity of the discussion and presentation. Very enjoyable reading and very lucid explanations of the main points. Almost always, when a question pops in the mind of the reader, it is answered in the subsequent lines.
3- Breadth and variety of the subjects addressed.
Weaknesses
1- Some arguments and some discussions are a bit qualitative. Mostly orders of magnitude are given. Often they have to be trusted, without much motivation.
2- References are not numerous. Perhaps a more extensive overview of work in these areas would have been welcome.
Report
These are very high quality, very entertaining and highly informative lecture notes, perfectly in line with their stated scope. They cover a wide range of slightly unconventional ideas, in a remarkably clear and coherent way.
If the precision and the quantitativeness of the discussion are sometimes a bit light, this probably reflects the fact that the notes are in "lecture style".
The notes can definitely be published, after the author has considered the small changes suggested below.
Requested changes
1- Page 1, typo: The self-interactions of the dark matter needs $\to$ need
2- Page 1: "there is less than O (1) scattering in the dark matter during the mergers of galaxy clusters": can the author please explain? O(1) scatterings in which time or volume or area?...
3- Page 9: please explicit the several acronyms used: NMR, ESR, SQUID, LC...
4- Page 9: typo: please use Larmor or larmor consistently in the two occurrences
5- Page 12: typo: sources of noise that is $\to$ are
6- Page 14, 7 lines from the bottom: please avoid some repetitions of "at low threshold"
7- Page 15: typo: The chemists also know to create... $\to$ know how to create...
8- Page 17: " This is because we are currently probing WIMP interactions with the standard model that are mediated by the higgs". This sentence is unclear, unless the reader knows what the authors has in mind. The author is referring to Direct Detection, and is probably thinking of diagrams with a higgs in the t-channel, which give elastic scattering cross-sections on nuclei in the ballpark of those currently probed by experiments. If this is the case, the author should make this more explicit, otherwise the logical short-circuit for a non-expert reader is not obvious (e.g. one could think of WIMP interactions in Indirect Detection, or at colliders, or in production mechanisms...).
9- Page 18: " With such directional detection capability, one can make use of the fact that, due to momentum conservation, when a solar neutrino collides with a nucleus, the recoiling nucleus has to move away from the Sun". All the discussion of directional detection is biased towards neutrinos from the Sun and therefore beating the neutrino floor, but directional detection is also useful to detect the DM wind. A note on this should be added.
10- Page 20: "absorbed by damaged part of the crystal" $\to$ "by the damaged part of the crystal" or "by damaged parts of the crystal" (?). Anyway please correct.
11- Page 20, 5 lines from the bottom: cm^2/gr $\to$ cm^2/g or gram, to respect the notations in the international system.
12- Page 21, first line: "likely" repetition
13- Page 22, 10 lines from the bottom: $g_\chi \chi^\star\chi$: is this missing a $\phi$?
14- Page 23: typo: 10-3 velocity $\to$ $10^{-3}$, and please make this explicit that this is in units of $c$
15- Page 23: and thus sensitive $\to$ and thus are sensitive
16- Page 25: "well below the Chandrasekhar limit": please explain shortly what the limit is and why it is relevant
17- Page 25: 10−2 is larger $\to$ $10^{−2}$ is larger, and please add units of $c$
18- Page 26: "which is $\gg$ keV ambient temperature": unclear, please revise.

---

## Round 2 · Author Response

I have incorporated the vast majority of the comments by the reviewers. The only comments I have not incorporated involve specifying the speed of light explicitly when I talk about velocities of dark matter since I think this is clear.

---

## Editorial Decision

published